**Data Availability Statement:** All data are accessible from the Figshare website at https:// figshare.com/articles/dataset/dataset/15245259.

# Effect of enterally administered sleep-promoting medication on the intravenous sedative dose and its safety and cost profile in mechanically ventilated patients: A retrospective cohort study

Takefumi Tsunemitsu[1]*, Yuki Kataoka[2], Masaru Matsumoto[1], Takashi Hashimoto[3], Takao Suzuki[1]

1 Department of Emergency and Critical Care Medicine, Hyogo Prefectural Amagasaki General Medical Center, Hyogo, Japan, 2 Hospital Care Research Unit, Hyogo Prefectural Amagasaki General Medical Center, Hyogo, Japan, 3 Department of Pharmacy, Hyogo Prefectural Amagasaki General Medical Center, Hyogo, Japan

* tsunemitsu0730@yahoo.co.jp

## Abstract

### Background

The clinical effect of enteral administration of sleep-promoting medication (SPM) in mechanically ventilated patients remains unclear. This study aimed to investigate the relationship between enteral SPM administration and the intravenous sedative dose and examine the safety and cost of enteral SPM administration.

### Methods

This single-center retrospective cohort study was conducted in a Japanese tertiary hospital intensive care unit (ICU). The exposure was enteral SPM administration during mechanical ventilation. The outcome was the average daily propofol dose per body weight administered as a continuous sedative during mechanical ventilation. Patients were divided into three groups based on the timing of SPM administration at ICU admission: "administration within 48 hours (early administration [EA])," "administration after 48 hours (late administration [LA])," and "no administration (NA)." We used multiple linear regression models.

### Results

Of 123 included patients, 37, 50, and 36 patients were assigned to the EA, LA, and NA groups, respectively. The average daily propofol dose per body weight was significantly lower in the EA group than in the LA and NA groups (β -5.13 [95% confidence interval (CI) -8.93 to -1.33] and β -4.51 [95% CI -8.59 to -0.43], respectively). Regarding safety, enteral SPM administration did not increase adverse events, including self-extubation. The total cost of neuroactive drugs tended to be lower in the EA group than in the LA and NA groups.

**Funding:** The authors received no specific funding for this work.

**Competing interests:** The authors have declared that no competing interests exist.

## Conclusions

Early enteral SPM administration reduced the average daily propofol dose per body weight without increasing adverse events.

## Introduction

Sleep disruption, which is often observed in critically ill patients admitted to the ICU [1–3], has negative effects on patients and can cause a worse prognosis. It causes abnormalities in immune and metabolic endocrine functions, which play crucial roles in critically ill patients [4, 5]. Additionally, it may be associated with the occurrence of delirium [6], noninvasive ventilation failure [7], and increased mortality [8]. Consequently, recent clinical guidelines have indicated the need for research methods to improve sleep [9].

Intravenous sedatives, including propofol and benzodiazepines, are administered to improve sleep efficiency [10]; however, they can negatively affect critically ill patients. Intravenous sedatives affect respiratory function, causing apnea and hypoxia, and circulation, causing bradycardia and hypotension [11–13]. Additionally, they may be associated with prolonged mechanical ventilation [14]. Moreover, intravenous sedatives tend to cause deep sedation [15], which is associated with a worse prognosis [16, 17].

Sleep-promoting medication (SPM), including melatonin, ramelteon, and atypical antipsychotics, may improve sleep efficiency in critically ill patients [9]. A pilot randomized controlled trial reported that melatonin may improve nocturnal sleep efficiency [18]. Regarding safety, several studies have reported safe enteral administration of neuroactive medications [19, 20].

To the best of our knowledge, no studies currently exist addressing the relationship between enteral SPM administration and the amount of intravenous sedatives, and the usefulness and safety of enteral SPM administration. This exploratory study aimed to investigate the relationship between enteral SPM administration and the intravenous sedative dose. Further, we aimed to examine the safety and cost of enteral SPM administration. We believe that this study is crucial to understand the effect of enteral SPM administration on mechanically ventilated patients.

## Materials and methods

### Study design

This single-center retrospective cohort study implied a revision of medical records. We obtained data from medical records of patients admitted to the ICU of Hyogo Prefectural Amagasaki General Medical Center. The study design and methodology were approved by the institutional review board. The requirement of written informed consent was waived given the retrospective design of the study. This study was reported following the Strengthening the Reporting of Observational studies in Epidemiology (STROBE) statement (Table in S1 File).

### Setting

This study was conducted in the emergency intensive care unit at a 700-bedded tertiary medical care center in Japan between July 2015 and January 2020. We started the study in July 2015 given that this was when our hospital opened.

## Patients

The inclusion criteria were as follows: aged $\geq$ 15 years, having started mechanical ventilation within 24 hours of ICU admission, and requiring mechanical ventilation using an oral endotracheal tube for at least 48 hours. Conversely, the exclusion criteria were central nervous system disease (e.g., stroke, epilepsy, meningitis, and encephalitis), cardiopulmonary arrest, traumatic brain injury, overdose, gastrointestinal tract impracticability (administration of continuous neuromuscular blockade, abdominal surgery, ileus, and gastrointestinal bleeding), previous psychiatric or cognitive pathology, Child C hepatopathy, pregnancy, previous tracheostomy, ICU readmission, lacking body weight records, death in ICU, and not receiving propofol.

## Exposure

Exposure was enteral SPM administration during mechanical ventilation through an oral endotracheal tube. This study defined the following medications as SPM: trazodone, mianserin, quetiapine, and suvorexant [21–23]. Exposure was considered as present if these drugs were administered after enteral nutrition in the evening or before sleep. We divided the patients into the following three groups: "administration within 48 hours of ICU admission (early administration [EA])," "administration after 48 hours of ICU admission (late administration [LA])," and "no administration [NA])." We set the cutoff at 48 hours since early sleep management appears to be more beneficial with consideration of sleep disruption pathophysiology [24] and enteral medication can be administered at the start of enteral feeding, which is recommended within 48 hours of ICU admission [25, 26]. SPM was administered at the discretion of the physician-in-charge. This study did not consider ramelteon and benzodiazepines as exposures. The reason we excluded ramelteon is that its short-term use has a weak sleep effect, and we thought it has little effect on sleep or sedation during ICU stay. In fact, a systematic review of ramelteon showed that although it significantly improved subjective sleep latency, the difference was only 4 minutes, and there was no difference in subjective total sleep time [27]. Moreover, melatonin, which is a drug similar to ramelteon, has insufficient evidence regarding its improvement of sleep quality [28] and is not strongly recommended by the European guidelines [22]. The use of benzodiazepines in the ICU has a high incidence of adverse events [29]. Other antipsychotics (e.g., haloperidol, risperidone, and yokukansan) were excluded from exposure since they were primarily administered for agitation and were rarely used for sleep improvement in our ICU.

## Outcomes

The primary outcome was the average daily propofol dose per body weight administered as a continuous sedative during mechanical ventilation through an oral endotracheal tube. The numerator was the total propofol dose administered as a continuous sedative during mechanical ventilation. In contrast, the denominator was the body weight and number of days of mechanical ventilation. We chose propofol since it is the most commonly used sedative in our ICU and its use is increasing worldwide [30]. A continuous sedative was defined as one administered for > 1 hour. The physician on duty set the target Richmond Agitation-Sedation Scale (RASS) for the sedative medication dose, with the nurses mainly taking initiative to increase or decrease it. There is no written protocol for sedation in our ICU. However, it was recommended that sedation vacations be performed during the daytime and that nighttime sedation be managed at RASS-3 to RASS-4.

Other outcomes included the duration of mechanical ventilation using an oral endotracheal tube, length of ICU stay, delirium diagnosed based on the Confusion Assessment Method for

the ICU or Intensive Care Delirium Screening Checklist, and agitation defined as an RASS ≧ 2 points during mechanical ventilation with oral endotracheal tube. We collected data regarding adverse events, including self-extubation, liver damage requiring drug change, unplanned sedative administration (haloperidol, hydroxyzine, and benzodiazepines), and ventricular arrhythmia. Other adverse events (e.g., self-removal of other invasive tools, anaphylaxis, hypotension, and paralytic ileus) could not be included in this study because it was difficult to extract reliable data due to the retrospective nature of the study. The cost of neuroactive drug was calculated by totaling the amount of each enteral and intravenous drug extracted in this study during the mechanical ventilation through an oral endotracheal tube based on the drug price in June 2020. The conversion from yen to dollar was calculated at 108 yen per dollar, which is the rate as of June 2020. Additionally, we collected data regarding the administration of analgesics (acetaminophen and fentanyl), other intravenous sedatives (midazolam and dexmedetomidine), and other neuroactive enteral drugs (ramelteon, yokukansan, perospirone, risperidone, and benzodiazepines).

## Statistical methods

Continuous data were described using medians with interquartile range and categorical data using frequencies and percentages. We constructed multiple linear regression models to examine the association between enteral SPM administration and the average daily propofol dose per body weight. We checked residuals in multiple liner regression for normal distribution using normal quantile-quantile plot. We set the following variables as covariates for the primary study outcome: age, sex, Charlson comorbidity index, pre-admission use of sleeping pills, diagnosis on ICU admission, maximum sequential organ failure assessment (SOFA) scores during the first week after admission, intravenous dexmedetomidine administration, intravenous midazolam administration, enteral ramelteon administration, enteral benzodiazepine administration, average daily fentanyl dose per body weight, and acetaminophen administration. For analysis of other outcomes, logistic regression analysis was used for binary variables while multiple linear regression models were used for continuous variables. We set the following variables as covariates for secondary study outcomes: age, sex, Charlson comorbidity index, maximum SOFA scores during the first week after admission, diagnosis on ICU admission, and intravenous midazolam administration. We did not use any special model to calculate the costs in this study. The cost was calculated by totaling the drugs used in this study. All tests were two-sided, with a significance level of 0.05. Analyses were performed using EZR (version 1.36) [31].

## Results

### Participants

As shown in Fig 1, we screened 1412 patients and included 123 patients for analysis. Table 1 shows the patient baseline characteristics. The median age of patients of this study was 72 years, 72% were males, and the median body mass index was 23.1. The SOFA score and Charlson comorbidity index were 6 and 0 in the EA group, 6 and 0 in the LA group, and 7 and 0 in the NA group. Respiratory failure was the most common diagnosis upon ICU admission, especially in the EA group (EA, 22 [60%]; LA, 28 [56%]; NA, 13 [36%]).

### Detail regarding sleep-promoting medication, sedatives, and analgesics

Table 2 shows data regarding drug use for analgesia, sedation, and sleep during mechanical ventilation through an oral endotracheal tube in each group. Trazodone was the most

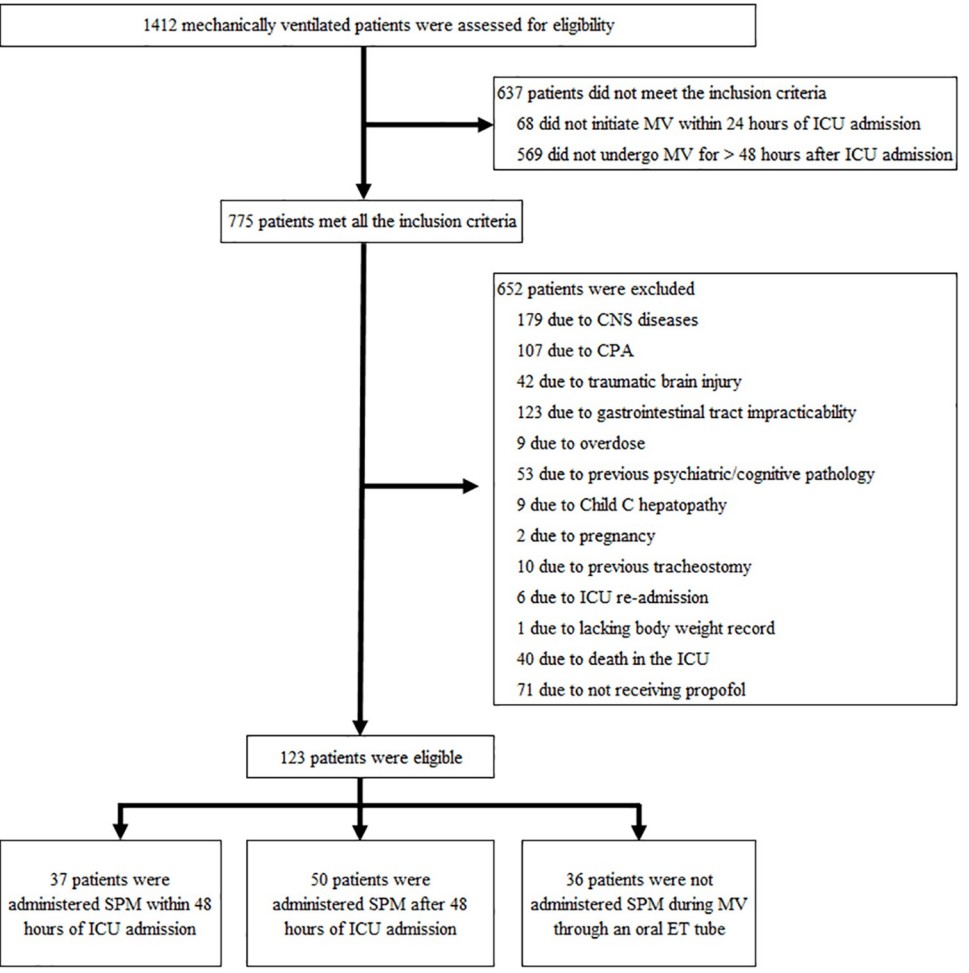

**Fig 1. Flow chart of study population.** MV, mechanical ventilation; CNS, central nervous system; CPA, cardiopulmonary arrest; ICU, intensive care unit; SPM, sleep-promoting medication; ET, endotracheal.

common drug administered as SPM (EA, 29 [78%]; LA, 40 [80%]; NA, 0 [0%]). The percentage administration of midazolam and dexmedetomidine was 41% and 27% in the EA group, 36% and 38% in the LA group, and 33% and 47% in the NA group, respectively. Regarding other enterally administered neuroactive drugs, ramelteon was administered in numerous patients in each group (EA, 34 [92%]; LA, 42 [84%]; NA, 25 [69%]). All the patients received fentanyl and the daily fentanyl dose were similar in each group (EA, 11.15 μg/kg/day [95% confidence interval (CI), 7.49–12.77], LA, 10 μg/kg/day [95% CI, 8.67–13.20], NA, 9.92 μg/kg/day [95% CI, 6.69–12.72]). The number of patients who received acetaminophen was 70% in the EA group, 58% in the LA group, and 39% in the NA group.

## Outcomes

Table 3 shows the study outcomes. In multivariate analysis, the average daily propofol dose per body weight administered as a continuous sedative during mechanical ventilation through an oral endotracheal tube was significantly lower in the EA group than in the LA and NA groups (β -5.13 [95% CI, -8.93 to -1.33], β -4.51 [95% CI, -8.59 to -0.43]).

**Table 1. Baseline characteristics.**

|  | All | EA group | LA group | NA group |
|---|---|---|---|---|
|  | (n = 123) | (n = 37) | (n = 50) | (n = 36) |
| Age, median [IQR], years | 72 [63–78] | 69 [58–78] | 71 [64–77] | 74 [66–79] |
| Male, n (%) | 88 (72) | 26 (70) | 40 (80) | 22 (61) |
| BMI, median [IQR] | 23.1 [20.0–25.6] | 23.1 [20.0–25.2] | 23.6 [20.1–26.4] | 23.1 [20.0–24.7] |
| SOFA score, median [IQR] | 6 [4–10] | 6 [4–10] | 6 [4–10] | 7 [5–9] |
| Charlson comorbidity index, median [IQR] | 0 [0–2] | 0 [0–2] | 0 [0–2] | 0 [0–2] |
| Diagnosis on ICU admission, n (%) |  |  |  |  |
| Respiratory failure | 63 (51) | 22 (60) | 28 (56) | 13 (36) |
| Cardiac failure | 21 (17) | 6 (16) | 5 (10) | 10 (28) |
| Sepsis | 18 (15) | 4 (11) | 8 (16) | 6 (17) |
| Trauma | 10 (8) | 3 (8) | 6 (12) | 1 (3) |
| Other | 11 (9) | 2 (5) | 3 (6) | 6 (17) |
| Pre-admission use of sleeping pills, n (%) | 10 (8) | 5 (14) | 2 (4) | 3 (8) |

SPM, sleep-promoting medication; ICU, intensive care unit; IQR, interquartile range; BMI, body mass index; SOFA, Sequential Organ Failure Assessment.

Multivariable analysis revealed no significant differences between groups in the duration of mechanical ventilation with oral endotracheal tube, length of ICU stay, RASS ≧ two points, and delirium (Table 3). Table in S2 File presents the results of univariate analysis of the primary and secondary outcomes. The results of univariate and multivariate analysis were similar.

**Table 2. Neuroactive drugs and analgesics during MV through an oral ET tube.**

|  | EA group | LA group | NA group |
|---|---|---|---|
|  | (n = 37) | (n = 50) | (n = 36) |
| Details of SPM, n (%) |  |  |  |
| Trazodone | 29 (78) | 40 (80) | 0 (0) |
| Quetiapine | 9 (24) | 6 (12) | 0 (0) |
| Mianserin | 9 (24) | 14 (28) | 0 (0) |
| Suvorexant | 2 (5.4) | 0 (0) | 0 (0) |
| Continuous IV neuroactive drugs |  |  |  |
| Midazolam, n (%) | 15 (41) | 18 (36) | 12 (33) |
| Average daily midazolam dose, mean [IQR], mg/kg/day | 0.13 [0.09–0.29] | 0.25 [0.14–1.04] | 0.23 [0.14–0.59] |
| Dexmedetomidine, n (%) | 10 (27) | 19 (38) | 17 (47) |
| Average daily dexmedetomidine dose, mean [IQR], μg/kg/day | 1.84 [0.76–3.09] | 2.06 [1.03–3.52] | 2.69 [0.73–4.55] |
| EN neuroactive drugs, n (%) |  |  |  |
| Ramelteon | 34 (92) | 42 (84) | 25 (69) |
| Yokukansan | 10 (27) | 17 (34) | 10 (28) |
| Benzodiazepines | 0 (0) | 7 (14) | 1 (3) |
| Other neuroactive drugs | 4 (11) | 6 (12) | 2 (6) |
| Analgesics |  |  |  |
| Fentanyl, n (%) | 37 (100) | 50 (100) | 36 (100) |
| Average daily fentanyl dose, mean [IQR], μg/kg/day | 11.15 [7.49–12.77] | 10.73 [8.67–13.20] | 9.92 [6.69–12.72] |
| Acetaminophen, n (%) | 26 (70) | 29 (58) | 14 (39) |
| Average daily acetaminophen dose, mean [IQR], mg/kg/day | 18.03 [12.03–25.53] | 12.70 [8.45–22.65] | 10.72 [6.21–25.59] |

MV, mechanical ventilation; ET, endotracheal; SPM, sleep-promoting medication; ICU, intensive care unit; IQR, interquartile range; IV, intravenous; EN, enteral.

**Table 3. Study outcomes.**

| | EA group (n=37) | LA group (n=50) | NA group (n=36) | EA group vs. LA group differences (95% CI) | p | EA group vs. NA group differences (95% CI) | p |
|---|---|---|---|---|---|---|---|
| Primary outcome[a] | | | | | | | |
| Average daily propofol dose, median [IQR],mg/kg/day | 2.1 [1.4 to 5.2] | 7.6 [2.1 to 13.3] | 6.9 [2.7 to 11.7] | β, -5.13 (-8.93 to -1.33) | <0.01 | β, -4.51 (-8.59 to -0.43) | 0.03 |
| Secondary outcome[b] | | | | | | | |
| MV duration through an oral ET tube, median [IQR], days | 6 [4 to 10] | 6 [5 to 9] | 6 [4 to 8] | β, -0.69 (-2.53 to 1.15) | 0.46 | β, 0.40 (-1.59 to 2.39) | 0.69 |
| Length of ICU stay, median [IQR], days | 7 [6 to 11] | 9 [7 to 13] | 9 [6 to 11] | β, -0.68 (-4.13 to 2.76) | 0.7 | β, -1.58 (-5.31 to 2.14) | 0.4 |
| RASS 2 points and more, n (%) | 15 (41) | 26 (52) | 18 (50) | OR, 0.56 (0.21 to 1.47) | 0.24 | OR, 0.76 (0.27 to 2.12) | 0.6 |
| Delirium, n (%) | 10 (27) | 17 (34) | 11 (31) | OR, 1.12 (0.39 to 3.18) | 0.84 | OR, 1.34 (0.43 to 4.20) | 0.61 |
| Adverse event, n (%) | | | | | | | |
| Self-removal of ET tube[c] | 0 (0) | 2 (4) | 2 (6) | RD, -4.00 (-7.21 to -0.79) | 0.02 | RD, -5.56 (-10.60 to -0.51) | 0.03 |
| Liver damage | 3 (8) | 9 (18) | 5 (14) | OR, 0.40 (0.10 to 1.60) | 0.2 | OR, 0.55 (0.12 to 2.48) | 0.43 |
| Ventricular arrhythmia | 2 (5) | 4 (8) | 4 (11) | OR, 0.66 (0.11 to 3.79) | 0.64 | OR, 0.46 (0.08 to 2.67) | 0.38 |
| Unplanned sedatives | 6 (16) | 11 (22) | 3 (8) | OR, 0.69 (0.23 to 2.06) | 0.5 | OR, 2.13 (0.49 to 9.26) | 0.31 |
| Daily cost, $/day | | | | | | | |
| Continuous IV sedatives[d] | 6.1 [3.2 to 14.1] | 14.1 [4.0 to 37.9] | 18.1 [8.9 to 36.3] | β, -9.8 (-22.0 to 2.4) | 0.11 | β, -17.6 (-30.8 to -4.5) | <0.01 |
| Bolus IV sedatives[e] | 0.0 [0.0 to 0.0] | 0.0 [0.0 to 0.0] | 0.0 [0.0 to 0.0] | β, 0.0 (-0.1 to 0.1) | 0.96 | β, 0.0 (-0.1 to 0.1) | 0.45 |
| EN sleep or sedative medication[f] | 0.9 [0.8 to 1.1] | 0.8 [0.4 to 0.9] | 0.5 [0.1 to 0.6] | β, 0.2 (0.1 to 0.4) | <0.01 | β, 0.5 (0.3 to 0.7) | <0.01 |
| AnalgesicsM[g] | 11.5 [9.8 to 15.3] | 12.1 [10.6 to 15.3] | 11.1 [8.5 to 12.4] | β, -0.8 (-3.0 to 1.3) | 0.46 | β, 0.2 (-2.1 to 2.5) | 0.88 |
| All neuroactive drugs | 19.5 [15.1 to 30.9] | 29.4 [18.3 to 50.3] | 30.3 [17.5 to 48.9] | β, -10.4 (-23.8 to 3.0) | 0.13 | β, -16.9 (-31.4 to -2.5) | 0.02 |

SPM, sleep-promoting medication; MV, mechanical ventilation; CI, confidence interval; IQR, interquartile range; β, β coefficient; ET, endotracheal; ICU, intensive care unit; RASS, Richmond Agitation-Sedation Scale; OR, odds ratio; RD, risk difference; IV, intravenous; EN, enteral.

a Multivariate analysis adjusted for age, sex, Charlson comorbidity index, pre-admission use of sleeping pills, diagnosis on ICU admission, maximum SOFA scores during the first week after admission, IV dexmedetomidine administration, IV midazolam administration, EN ramelteon administration, EN benzodiazepines administration, average daily fentanyl dose per body weight, and acetaminophen administration.

b Multivariate analysis adjusted for age, sex, Charlson comorbidity index, maximum SOFA scores during the first week after admission, diagnosis on ICU admission, and IV midazolam administration.

c Since we could not calculate the odds ratio due to the zero cell, we assessed through the risk difference. Risk difference is described as a percentage.

d Propofol, midazolam and dexmedetomidine were included in the calculation.

e Haloperidol, hydroxyzine, benzodiazepines, and propofol which were not administered as continuous sedatives were included in the calculation.

f Trazodone, quetiapine, mianserin, suvorexant, ramelteon, yokukansan, perospirone, risperidone, and benzodiazepines were included in the calculation.

g Fentanyl and acetaminophen were included in the calculation.

As a sensitivity analysis, we compared patients who received SPM during mechanical ventilation with those who did not, and found no difference in primary or secondary outcomes between the two groups, as shown in Table in S3 File.

Table 3 shows the incidence of adverse events. There were no differences between groups in liver damage requiring drug change, ventricular arrhythmia, or unplanned sedative administration. The risk difference of self-extubation were lower in the EA group than in the EA and LA groups (risk difference [RD] -4.00%, p = 0.02; EA vs. NA, RD -5.56%, p = 0.03). Since no patients in the EA group had self-extubation and we could not calculate the odds ratios, self-extubation was assessed through risk difference.

In the EA group, the daily cost for enterally administered sleep or sedative medicine administered was significantly higher than in the LA and NA groups (β 0.2 [95% CI, 0.1–0.4], β 0.5 [95% CI, 0.3–0.7]); contrastingly, the daily cost for continuous intravenous sedatives was

significantly lower in the EA group than in the NA group (β -17.6 [95% CI, -30.8 to -4.5]). For all neuroactive drugs (SPM, sedatives, and analgesics), the daily cost was significantly lower in the EA group than in the NA group (β -16.9 [95% CI -31.4 to -2.5]).

## Discussion

Our findings indicated that the EA group had significantly lower propofol doses than the LA and NA groups. Further, there were no differences between groups in the duration of mechanical ventilation, length of ICU stay, RASS > 2 points, and delirium incidence. There was no difference between groups in adverse events except for self-extubation, which was less common in the EA group. Although the EA group showed significantly increased costs for enteral sleep and sedative medication compared to other groups, it showed significantly reduced costs for continuous intravenous sedatives and all neuroactive drugs compared to the NA group.

Early enteral SPM administration is an alternative to intravenous sedatives during nighttime sleep. In our study, the EA group showed a reduced propofol dose, which could be attributed to reduced propofol administered for nighttime sleep. Propofol is commonly administered in the ICU for nighttime sleep [10]; however, it is known to alter sleep structure [32]. Moreover, clinical guidelines suggest that it should not be administered for sleep improvement [9]. The Cochrane review reported lacking evidence for propofol administration for sleep efficiency improvement since previous studies are highly heterogeneous in terms of design, comparative agents, and participant groups; further, several biases are present [33]. Enteral SPM administration may have contributed to nighttime sleep and reduced the propofol dose at night.

Notably, the average daily propofol dose was similar in the LA and NA groups. In addition, sensitivity analysis showed that there was no difference in the average daily propofol dose in the SPM group (EA group plus LA group) compared to the non-SPM group, suggesting that a delay in SPM administration may have less effect on the average daily propofol dose. Although it is impossible to make a definitive conclusion due to the unmeasured confounding factors, in case of sleep disruption and sleep rhythm disturbances caused by delayed sleep intervention, sleep medications may be less effective. Therefore, we believe that early SPM administration may be desirable; however, the appropriate timing of SPM administration remains unclear.

Enteral SPM administration did not cause excessive light sedation or increase adverse events. Although the EA group had the lowest propofol dose, there was no increase in the number of patients with an RASS ≧ 2 points, self-extubation rate, and unplanned sedative administration. Therefore, SPM administration could not have resulted in excessive light sedation caused by reducing the propofol dose. Regarding the safety of enteral SPM administration, there was no among-group difference in occurrence of liver damage requiring a change in medication or ventricular arrhythmia caused by QT prolongation. These results suggest that enteral SPM administration may not increase such adverse events. However, the small sample size could have impeded achievement of a statistical difference. Additionally, routine SPM administration is not recommended in clinical guidelines due to poor evidence [9]; therefore, future studies should further examine its safety.

Regarding the costs for all neuroactive drugs, the EA group was significantly more advantageous than the NA group. Although enteral SPM administration increased the cost of enteral medication, it decreased the intravenous sedative dose, which is more expensive, and consequently lowered the total costs. Previous findings on comparisons of enteral and intravenous sedative administration have been consistent [34]. Generally, enteral medications are cheaper than intravenous medications; therefore, enteral drug administration is cost-effective. If the

intestinal tract is available, it may be more cost-effective to enterally administer sleep and sedative drugs.

This study had several limitations. First, this was a single-center study with a small sample size, which may underestimate the significance of the outcomes. Second, we could not adjust for some potential covariates given that this was a retrospective study with limited information. Third, since we did not examine for actual sleep quality improvement, it is unclear whether sleep improvement resulting from SPM administration affected the outcomes. Additionally, some sedative drugs were administered as SPMs and their sedative effect could have affected the outcomes. Although it is difficult to separate sleep from sedation in clinical practice, enterally administered SPMs were only used during the nighttime period. Therefore, the SPM effect on sedation was thought to only occur during the nighttime period and its impact on the results may be negligible. Fourth, the choice of the SPM was based on the authors' own experience, and it is unclear whether other drugs would have the same efficacy. To validate the results of this study, randomized control trials need to be conducted with the drugs used in this study as exposures.

## Conclusions

We found that early enteral administration of SPM reduced the average daily propofol dose per body weight without increasing adverse events; moreover, it had a cost advantage. There is a need for further prospective studies to confirm our findings and evaluate the relationship between SPM administration and other clinical outcomes.

## Supporting information

**S1 File. STROBE statement—checklist of items that should be included in reports of *cohort studies*.**
(DOCX)

**S2 File. Univariate analysis of study outcomes.**
(DOCX)

**S3 File. Comparison between all patients receiving and not receiving SPM.**
(DOCX)

**S4 File. Normal Q-Q plot.**
(DOCX)

**S5 File. TREND checklist.**
(PDF)

**S6 File. Study protocol in Japanese.**
(DOCX)

**S7 File. Study protocol in English.**
(DOCX)

## Acknowledgments

We would like to thank Editage (www.editage.com) for English language editing.

## Author Contributions

**Conceptualization:** Takefumi Tsunemitsu.

**Data curation:** Takefumi Tsunemitsu.

**Formal analysis:** Takefumi Tsunemitsu.

**Investigation:** Takefumi Tsunemitsu.

**Methodology:** Takefumi Tsunemitsu, Yuki Kataoka, Takashi Hashimoto.

**Project administration:** Takefumi Tsunemitsu.

**Validation:** Masaru Matsumoto, Takao Suzuki.

**Writing – original draft:** Takefumi Tsunemitsu.

**Writing – review & editing:** Yuki Kataoka, Masaru Matsumoto, Takao Suzuki.

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
