## [Decision Letter · Decision Letter 0]

20 Jul 2021

PONE-D-21-05291

Effect of enterally administered sleep-promoting medication on the intravenous sedative dose and its safety and cost profile in mechanically ventilated patients: a retrospective cohort study

PLOS ONE

Dear Dr. Tsunemitsu,

Thank you for submitting your manuscript to PLOS ONE. After careful consideration, we feel that it has merit but does not fully meet PLOS ONE’s publication criteria as it currently stands. Therefore, we invite you to submit a revised version of the manuscript that addresses the points raised during the review process.

In particular, statistical methods need to be revised with the help of a statistician in order to address all Reviewers' concerns.

We look forward to receiving your revised manuscript.

Kind regards,

Laura Pasin

Academic Editor

PLOS ONE

Journal Requirements:

2. In the ethics statement in the manuscript and in the online submission form, please provide additional information about the patient records/samples used in your retrospective study, including the source of the medical records/samples analyzed in this work (e.g. hospital, institution or medical center name).

3. We noted that you submitted this study as a clinical trial, but according to your description and the WHO definition of clinical trials we would not consider this a clinical trial but rather a retrospective observational study. In order to avoid confusion we would suggest that you change the wording in your manuscript and avoid referring to this study as a clinical trial.

5. Please include captions for ALL your Supporting Information files at the end of your manuscript, and update any in-text citations to match accordingly. Please see our Supporting Information guidelines for more information: http://journals.plos.org/plosone/s/supporting-information.

Reviewers' comments:

Reviewer's Responses to Questions

**Comments to the Author**

1. Is the manuscript technically sound, and do the data support the conclusions?

Reviewer #1: No

Reviewer #2: Yes

Reviewer #3: Yes

2. Has the statistical analysis been performed appropriately and rigorously? 

Reviewer #1: No

Reviewer #2: Yes

Reviewer #3: Yes

3. Have the authors made all data underlying the findings in their manuscript fully available?

Reviewer #1: No

Reviewer #2: Yes

Reviewer #3: No

4. Is the manuscript presented in an intelligible fashion and written in standard English?

Reviewer #1: No

Reviewer #2: Yes

Reviewer #3: No

5. Review Comments to the Author

Reviewer #1: I have no competing interest to declare. I declare that I did not make use of any material or take advantage of any information I gained through the peer review process.

GENERAL COMMENT

Dr. Tsunemitsu et al. performed a single-center retrospective cohort study to assess the effect of enteral sleep-promoting medications (SPM) in invasively mechanically ventilated adults on the dosages of intravenously administered propofol, adverse effects, and costs. They observed that early enteral administration of SPM reduced the average daily propofol dose per body weight without increasing adverse events (in fact, lower auto-extubation occurrence in the early administration group than in the late and no-SPM groups was reported) and was associated with reduced cost of continuous intravenous sedatives and all neuroactive drugs compared with no SPM.

The topic is relevant for daily clinical practice and deserve attention. However, I would have some major concerns related to the methodology the authors employed.

The 48-hour threshold for differentiating early from late SPM administration seems somewhat arbitrary (despite the cited references) and the authors should provide a comparison between all patients receiving SPM as a whole and patients not receiving them, at least as sensitivity analysis.

Moreover, although I am not a statistician, I do not think the intervention groups should be compared with pairwise comparisons like the authors did in their multiple regression models. I think that treating the categorical independent variable (timing of SPM) in the models by expressing it as 2 dichotomous dummy variables with the third as reference (e.g., LA and NA with EA as reference) and providing the comparison between the three groups together would prevent the increase in the type 1 error that may ensue from the statistical analysis that was performed in the study. I would recommend the authors having the methods reviewed by a statistician.

In addition, the choice of the SPMs was based on the authors’ own experience. This should be emphasized in the limitations section.

Finally, I would suggest the authors having the entire manuscript edited by an English-speaking editor due to some grammatical errors and potential language revisions (examples: “Intravenous sedatives affect respiratory function, which causes apnea and hypoxia, and circulation, which causes bradycardia and hypotension”; “We chose to start the study on July 2015 given that this was when our hospital opened.”; “The inclusion criteria were as followed”; “which is recommended as within 48 hours of ICU admission”; “Propofol was chosen since it is the most commonly used sedative in our ICU and its increasing worldwide usage”; “central nerve system”; “were administered SPM” in Fig. 1).

ABSTRACT

• “The average daily propofol dose per body weight was significantly lower in the EA group than in the LA and NA groups (β -5.18 [95% confidence interval (CI) -8.95 to -1.40], β -4.51 [95% CI -8.58 to -0.44]).” I would add “respectively” at the end of the sentence.

• “Moreover, it had a cost advantage.” The data reported in the abstract do not seem to substantiate this conclusion if there is only a trend towards reduced costs in the EA group (“The total cost of neuroactive drugs tended to be lower in the EA group than in the LA and NA groups”).

INTRODUCTION

• “Enteral SPM administration may be safely feasible for critically ill patients”. This sentence looks like a repetition of the previous one. Please keep only one of them.

• “We hypothesized that enteral SPM administration promotes sleep and reduces the intravenous sedative dose administered for sleep”. I would recommend the authors to clarify what they mean by sleep promotion and to consider removing this hypothesis from the introduction, since they could not assess sleep quality but only propofol doses. The reduction in intravenous sedative doses does not imply an improved quality of sleep. Indeed, propofol may not improve sleep quality in critically ill patients.

MATERIALS AND METHODS

• “This is because short-term ramelteon administration shows significant improvements in sleep; however, the differences were small [27]”. Why should a small difference in sleep improvement associated with ramelteon prevent its inclusion as SPM in the study? Please clarify.

• “The physician on duty set the target Richmond Agitation-Sedation Scale (RASS) for the sedative medication dose”. Was there a protocol in place for analgesia and sedation in the authors’ ICU with a target RASS?

• “We collected data regarding adverse events, including self-extubation, liver damage requiring drug change, unplanned sedative administration (haloperidol, hydroxyzine, and benzodiazepines), and ventricular arrhythmia.” The authors should better explain the rationale for including these adverse events and not others, otherwise their choice would seem somewhat discretional.

• “The cost was calculated by totaling the amount of each enteral and intravenous drug”. I would specify that the authors are speaking of sedative drugs.

• The authors should specify why they did not include yokukansan as exposure (similarly to ramelteon and benzodiazepines).

• See the comment of the statistical analysis in the general comment above. I think the comparison between the three groups according to baseline characteristics (Table 1) and neuroactive drugs and analgesics (Table 2) should be best performed with ANOVA. However, no statistical test was apparently performed. I would suggest to consult a statistician to get more insight into the statistical analysis.

• I understand the higher number of covariates used for the linear regression models for the primary outcome that the secondary outcomes. However, some covariates used for the secondary outcomes should be applied for the primary outcome as well, e.g., Charlson comorbidity index. I would suggest using maximum SOFA score during the first week after admission also for secondary outcomes, unless this is what the authors mean by “SOFA scores for one week after admission”.

RESULTS

• I would recommend consistency between the terms used in the text (EA, LA, NA) and those used in the tables (“SPM within 48 hours”, “SPM after 48 hours”, and “No SPM”).

• “There were no among-group differences in the age, sex, body mass index, SOFA scores on admission, Charlson comorbidity index, and use of sleeping pills before ICU admission.” P-values from an appropriate statistical test should be reported in Table 1 in a non-RCT study.

• “There was no among-group difference in the percentage administration of midazolam and dexmedetomidine as continuous sedatives.”; “Midazolam and dexmedetomidine were most commonly administered in the EA (EA: 15 [41%], LA: 18 [36%], NA: 12 [33%]) and NA groups”; “The number of patients who received acetaminophen was highest in the EA group (EA: 26 [70%], LA: 29 [58%], NA: 14 [39%]).”: these sentences are not justified by the data presented in Table 2, since p-values are missing. P-values from an appropriate statistical test should be reported.

• The statistical analysis may need to be reviewed with the help of a statistician. However, if the authors continue comparing two group at a time, they cannot use “among-group” but “between-group”. The same is true for the discussion.

• “contrastingly , the daily cost for continuous intravenous sedatives tended to be lower”. The cost for continuous intravenous sedatives was significantly lower for the comparison EA vs. NA, if I am not mistaken. I would not highlight results that tend towards statistical significance, e.g., cost for continuous intravenous sedatives in the comparison EA vs. LA, but only significant results.

• “For all neuroactive drugs (SPM, sedatives, and analgesics), the daily costs tended to be lower in the EA group than in the LA and NA groups.” See previous comment.

TABLE 1

• “Mechanical ventilation” does not seem to be reported in the table and should be removed from the caption.

• P-values should be reported to test each qualitative or quantitative variable difference among the groups.

TABLE 2

• P-values should be reported to test each qualitative or quantitative variable difference among the groups.

TABLE 3

• “MV through an oral ET tube”: I would change to “MV duration through an oral ET tube”.

DISCUSSION

• “Regarding the costs for all neuroactive drugs, the EA group showed an advantageous tendency compared with the other groups”. The trend could be observed for the EA vs. LA comparison, but it passed the threshold for statistical significance for the EA vs. NA comparison. Provided that statistical analysis may need to be reviewed, this should be accounted for in the discussion and conclusions.

• The choice of SPM was discretional and based on the authors’ own experience. This should be emphasized in the limitations section.

SUPPLEMENTARY

Supplementary materials should all be referred to in the main text.

Reviewer #2: Tsunemitsu and coworkers performed a retrospective review investigating how sleep-promoting medication could reduce intravenous sedative dose in mechanically ventilated patients.

They reviewed 5-years data of a single center, selecting 123 patients out of the 1412 admitted to the ICU during the study period, according to a preplanned study protocol. Administration of razodone, mianserin, quetiapine, and suvorexant within 48 hours of ICU admission resulted in lower average daily propofol doses - which was the study primary outcome. Moreover, the authors also report a lower incidence of self removal of ET tube in patients receiving sleep promoting medications.

Although the study in retrospective, single center and rather small, it addresses an interesting topic in a timely manner, and may therefore be useful to guide clinical practice.

The authors correctly acknowledge this limitations within the paper.

I would only suggest some corrections to improve fluency.

Here are a few examples:

Introduction

- Second paragraph: ARE administered. Please change

- Second paragraph: Please change “which cause” to “causing” in both instances to improve readability

- “Currently, there have been no studies on”. Please change to “To the best of our knowledge, no studies currently exist addressing…”

Methods

- Study design: “…study employed medical record information “. Please changed to “implied a revision of medical records”

Reviewer #3: A retrospective cohort study of patients divided into three groups was conducted to assess the clinical effect of enteral administration of a sleep-promoting medication (SPM) compared to intravenous sedative dose and assess the safety and cost of enteral SPM. The daily propofol dose per body weight was significantly lower in the early administration group compared to the other two groups. The rate of adverse events was not increased with enteral SPM administration.

Minor revisions:

1- Clarify if patients were divided randomly into the three groups.

2- Indicate if the data was checked for normal distribution or transformed to a normal distribution when using multiple linear regression.

3- Page 7: Indicate the statistical testing method(s) used to conclude, “There were no among-group differences . . .”

4- Table 1: Replace “man” with “male.”

5- In the statistical methods section, clarify the model used for summarizing the average daily cost.

6- Add line numbers to assist in the review process.

7- Carefully proofread the document. Use complete sentences and check grammar.

8- State and justify the study’s target sample size with a pre-study statistical power calculation. The power calculation should include: (1) the estimated outcomes in each group; (2) the α (type I) error level; (3) the statistical power (or the β (type II) error level); (4) the target sample size and (5) the statistical testing method and (6) for continuous outcomes, the standard deviation of the measurements.

6. PLOS authors have the option to publish the peer review history of their article (what does this mean?). If published, this will include your full peer review and any attached files.

Reviewer #1: No

Reviewer #2: **Yes: **Pasquale Nardelli

Reviewer #3: No

---

## [Author Response · Author response to Decision Letter 0]

18 Aug 2021

Emily Chenette

Editor-in-Chief

PLOS ONE 

Dear Editor,

Manuscript No: PONE-D-21-05291

Title: Effect of enterally administered sleep-promoting medication on the intravenous sedative dose and its safety and cost profile in mechanically ventilated patients: A retrospective cohort study

We would like to express our sincere thanks to you and the reviewers for the thorough review of our manuscript and for the opportunity to submit a revised and improved version. We have carefully reviewed the comments and revised the manuscript on the basis of the reviewers’ comments. Our point-by-point responses to the reviewers’ comments are listed below. 

Journal Requirements:

Response: We have modified our manuscript to meet the requirements.

2. In the ethics statement in the manuscript and in the online submission form, please provide additional information about the patient records/samples used in your retrospective study, including the source of the medical records/samples analyzed in this work (e.g. hospital, institution or medical center name).

Response: We have added the following sentence in the section stating the study design: “We obtained data from medical records of patients admitted to the ICU of Hyogo Prefectural Amagasaki General Medical Center.” 

3. We noted that you submitted this study as a clinical trial, but according to your description and the WHO definition of clinical trials we would not consider this a clinical trial but rather a retrospective observational study. In order to avoid confusion we would suggest that you change the wording in your manuscript and avoid referring to this study as a clinical trial.

Response: We have resubmitted this paper as a research paper, not as a clinical trial, and made revised relevant text in the manuscript accordingly.

4. We note that you have indicated that data from this study are available upon request. PLOS only allows data to be available upon request if there are legal or ethical restrictions on sharing data publicly. 

Response: All data are accessible from the Figshare website at https://figshare.com/articles/dataset/dataset/15245259. 

5. Please include captions for ALL your Supporting Information files at the end of your manuscript, and update any in-text citations to match accordingly.

Response: We have rewritten supporting information citation in the manuscript according to the Supporting Information guidelines. In the section of study design in Materials and Methods, we have changed “S1 Table” to “Table in S1 File”, and “S2 Table” to “Table in S2 File” in the Outcomes of Results section.

Reviewer #1

・The 48-hour threshold for differentiating early from late SPM administration seems somewhat arbitrary (despite the cited references) and the authors should provide a comparison between all patients receiving SPM as a whole and patients not receiving them, at least as sensitivity analysis.

Response: We agree with the reviewer’s concerns. We performed sensitivity analysis comparing all patients receiving SPM with patients not receiving SPM. We have added a new table in the file named “S3 File”. Moreover, we have added the following sentence in Outcomes in the Results section: “As a sensitivity analysis, we compared patients who received SPM with those who did not and found no difference in primary or secondary outcomes between the two groups, as shown in Table in S3 File.”

・Moreover, although I am not a statistician, I do not think the intervention groups should be compared with pairwise comparisons like the authors did in their multiple regression models. I think that treating the categorical independent variable (timing of SPM) in the models by expressing it as 2 dichotomous dummy variables with the third as reference (e.g., LA and NA with EA as reference) and providing the comparison between the three groups together would prevent the increase in the type 1 error that may ensue from the statistical analysis that was performed in the study. I would recommend the authors having the methods reviewed by a statistician.

Response: We understand the reviewer’s concern. In contrast, the problem of multiple testing is unavoidable due to the exploratory nature of the research question. We consulted a statistician and decided to present limited two comparisons (EA as reference). We have removed the column “SPM after 48 hours vs. no SPM” in Table 3.

We have decided not to compare the NA and LA groups; hence, we have rewritten the relevant sentence in the Discussion section: “Notably, there was no difference in the primary outcome between the LA and NA groups, which suggests that delayed SPM administration may be ineffective in reducing the propofol dose. We expected that propofol doses in the LA group would be lower than those in the NA group. However, even after multivariate analysis that included illness severity and administration of other sedatives, there was no difference in the administered propofol doses between the LA and NA groups.” to “Notably, the average daily propofol dose was similar in the LA and NA groups. In addition, sensitivity analysis showed that there was no difference in the average daily propofol dose in the SPM group (EA group plus LA group) compared to the non-SPM group, suggesting that a delay in SPM administration may have less effect on the average daily propofol dose.”

・In addition, the choice of the SPMs was based on the authors’ own experience. This should be emphasized in the limitations section.

Response: We thank the reviewer for providing these insights. We have added the following sentence in the limitations section: “Fourth, the choice of the SPM was based on the authors’ own experience, and it is unclear whether other drugs would have the same efficacy. To validate the results of this study, randomized control trials need to be conducted with the drugs used in this study as exposures.”

・Finally, I would suggest the authors having the entire manuscript edited by an English-speaking editor due to some grammatical errors and potential language revisions (examples: “Intravenous sedatives affect respiratory function, which causes apnea and hypoxia, and circulation, which causes bradycardia and hypotension”; “We chose to start the study on July 2015 given that this was when our hospital opened.”; “The inclusion criteria were as followed”; “which is recommended as within 48 hours of ICU admission”; “Propofol was chosen since it is the most commonly used sedative in our ICU and its increasing worldwide usage”; “central nerve system”; “were administered SPM” in Fig. 1).

Response: We have enlisted the services of Editage (www.editage.com) to proofread the revised manuscript, along with our responses to the reviewers’ comments.

ABSTRACT

・“The average daily propofol dose per body weight was significantly lower in the EA group than in the LA and NA groups (β -5.18 [95% confidence interval (CI) -8.95 to -1.40], β -4.51 [95% CI -8.58 to -0.44]).” I would add “respectively” at the end of the sentence.

Response: We have rewritten the sentence according to the reviewer’s comment.

・“Moreover, it had a cost advantage.” The data reported in the abstract do not seem to substantiate this conclusion if there is only a trend towards reduced costs in the EA group (“The total cost of neuroactive drugs tended to be lower in the EA group than in the LA and NA groups”).

Response: We have removed the sentence “Moreover, it had a cost advantage.”

INTRODUCTION

・“Enteral SPM administration may be safely feasible for critically ill patients”. This sentence looks like a repetition of the previous one. Please keep only one of them.

Response: We have removed the sentence. 

・“We hypothesized that enteral SPM administration promotes sleep and reduces the intravenous sedative dose administered for sleep”. I would recommend the authors to clarify what they mean by sleep promotion and to consider removing this hypothesis from the introduction, since they could not assess sleep quality but only propofol doses. The reduction in intravenous sedative doses does not imply an improved quality of sleep. Indeed, propofol may not improve sleep quality in critically ill patients.

Response: We agree with the reviewer’s assessment. We have removed the sentence “We hypothesized that enteral SPM administration promotes sleep and reduces the intravenous sedative dose administered for sleep”.

MATERIALS AND METHODS

・“This is because short-term ramelteon administration shows significant improvements in sleep; however, the differences were small [27]”. Why should a small difference in sleep improvement associated with ramelteon prevent its inclusion as SPM in the study? Please clarify.

Response: We thank the reviewer for providing these insights. We have rewritten the sentences according to the reviewer’s comments: “This is because short-term ramelteon administration shows significant improvements in sleep; however, the differences were small [27]” to “The reason we excluded ramelteon is that its short-term use has a weak sleep effect and we thought it has little effect on sleep or sedation during ICU stay. In fact, a systematic review of ramelteon showed that although it significantly improved subjective sleep latency, the difference was only 4 minutes, and there was no difference in subjective total sleep time [27].”

・“The physician on duty set the target Richmond Agitation-Sedation Scale (RASS) for the sedative medication dose”. Was there a protocol in place for analgesia and sedation in the authors’ ICU with a target RASS?

Response: We thank the reviewer for providing these insights. We have added the following sentence in the Outcomes section: “There is no written protocol for sedation in our ICU. However, it was recommended that physicians should perform sedation vacations during the daytime and manage nighttime sedation at RASS-3 to RASS-4.”

・“We collected data regarding adverse events, including self-extubation, liver damage requiring drug change, unplanned sedative administration (haloperidol, hydroxyzine, and benzodiazepines), and ventricular arrhythmia.” The authors should better explain the rationale for including these adverse events and not others, otherwise their choice would seem somewhat discretional.

Response: We thank the reviewer for providing these insights. As adverse events were associated with inadequate sedation, we considered self-extubation, self-removal of other invasive tools, and unplanned sedative administration based on previous studies [1]. Moreover, we considered adverse events of the drug itself, such as liver damage, arrhythmia, hypotension, paralytic ileus, and anaphylaxis, referring to the drug information. However, due to the nature of the study, we were only able to extract reliable data on the four items shown in this study. We have incorporated the reviewer’s comments by adding the following sentence in the Outcomes section: “Other adverse events (e.g., self-removal of other invasive tools, anaphylaxis, hypotension, and paralytic ileus) could not be included in this study because it was difficult to extract reliable data due to the retrospective nature of the study.”

[1] Crit Care. 2019 Jan 7;23(1):3. doi: 10.1186/s13054-018-2280-x. 

・“The cost was calculated by totaling the amount of each enteral and intravenous drug”. I would specify that the authors are speaking of sedative drugs.

Response: We agree with the reviewer’s assessment. We have changed “The cost was calculated by totaling the amount of each enteral and intravenous drug” to “The cost of neuroactive drug was calculated by totaling the amount of each enteral and intravenous drug extracted in this study during the mechanical ventilation through an oral endotracheal tube based on the drug price in June 2020.”

・The authors should specify why they did not include yokukansan as exposure (similarly to ramelteon and benzodiazepines).

Response: The reviewer has raised in important question. To our knowledge, there are no RCTs on the sleep effects of yokukansan, and its effects on sleep are unknown. In addition, we do not actually use yokukansan as an SPM in our ICU. Thus, we excluded it as exposure. We have incorporated the reviewer’s comments by revising the relevant phrase in the section on exposure: “Other antipsychotics (haloperidol, risperidone, etc.)” to “Other antipsychotics (e.g., haloperidol, risperidone, and yokukansan).”

・See the comment of the statistical analysis in the general comment above. I think the comparison between the three groups according to baseline characteristics (Table 1) and neuroactive drugs and analgesics (Table 2) should be best performed with ANOVA. However, no statistical test was apparently performed. I would suggest to consult a statistician to get more insight into the statistical analysis.

Response: We appreciate the opportunity to discuss and clarify this point. We believe that comparing baseline characteristics and status of administration of neuroactive and analgesic drugs does not make much sense and causes increase in type 1 error [1,2]. Hence, we decided not to include p values in Table 1 and Table 2.

[1] Vandenbroucke JP, et al; STROBE Initiative. Strengthening the Reporting of Observational Studies in Epidemiology (STROBE): explanation and elaboration. PLoS Med. 2007 Oct 16;4(10):e297. doi: 10.1371/journal.pmed.0040297. 

[2] Palesch YY. Some common misperceptions about P values. Stroke. 2014;45(12):e244-e246. doi:10.1161/STROKEAHA.114.006138

・I understand the higher number of covariates used for the linear regression models for the primary outcome that the secondary outcomes. However, some covariates used for the secondary outcomes should be applied for the primary outcome as well, e.g., Charlson comorbidity index. I would suggest using maximum SOFA score during the first week after admission also for secondary outcomes, unless this is what the authors mean by “SOFA scores for one week after admission”.

Response: We have applied the Charlson comorbidity index for primary outcome as a covariate. Further, we have changed all “maximum SOFA scores for one week after admission” to “maximum SOFA score during the first week after admission”.

RESULTS

・I would recommend consistency between the terms used in the text (EA, LA, NA) and those used in the tables (“SPM within 48 hours”, “SPM after 48 hours”, and “No SPM”).

Response: We have changed “SPM within 48 hours” to “EA group,” “SPM after 48 hours” to “LA group,” and “No SPM” to “NA group.”

・“There were no among-group differences in the age, sex, body mass index, SOFA scores on admission, Charlson comorbidity index, and use of sleeping pills before ICU admission.” P-values from an appropriate statistical test should be reported in Table 1 in a non-RCT study.

Response: The reviewer has raised a significant point. As mentioned above, we decided not to test for differences in patient background; thus, we have revised the sentence to avoid misinterpretation.

We have changed the sentence to “The median age of patients of this study was 72 years, 72% were males, and the median body mass index was 23.1. The SOFA score and Charlson comorbidity index were similar in each group.”

・“There was no among-group difference in the percentage administration of midazolam and dexmedetomidine as continuous sedatives.”; “Midazolam and dexmedetomidine were most commonly administered in the EA (EA: 15 [41%], LA: 18 [36%], NA: 12 [33%]) and NA groups”; “The number of patients who received acetaminophen was highest in the EA group (EA: 26 [70%], LA: 29 [58%], NA: 14 [39%]).”: these sentences are not justified by the data presented in Table 2, since p-values are missing. P-values from an appropriate statistical test should be reported.

Response: 

The reviewer raised a significant point; however, as previously mentioned, we decided not to include P-values. 

We replaced the sentences “There was no among-group difference in the percentage administration of midazolam and dexmedetomidine as continuous sedatives” and “Midazolam and dexmedetomidine were most commonly administered in the EA (EA: 15 [41%], LA: 18 [36%], NA: 12 [33%]) and NA groups” with “The percentage administration of midazolam and dexmedetomidine was 41% and 27% in the EA group, 36% and 38% in the LA group, and 33% and 47% in the NA group, respectively.”

Further, we have changed “The number of patients who received acetaminophen was highest in the EA group (EA: 26 [70%], LA: 29 [58%], NA: 14 [39%]).” to “The number of patients who received acetaminophen was 70% in the EA group, 58% in the LA group, and 39% in the NA group.”

・The statistical analysis may need to be reviewed with the help of a statistician. However, if the authors continue comparing two group at a time, they cannot use “among-group” but “between-group”. The same is true for the discussion.

Response: We agree with you and have incorporated this suggestion throughout our paper.

・“contrastingly , the daily cost for continuous intravenous sedatives tended to be lower”. The cost for continuous intravenous sedatives was significantly lower for the comparison EA vs. NA, if I am not mistaken. I would not highlight results that tend towards statistical significance, e.g., cost for continuous intravenous sedatives in the comparison EA vs. LA, but only significant results.

Response: We thank the reviewer for providing these insights. We have changed the sentence “contrastingly, the daily cost for continuous intravenous sedatives tended to be lower” to “contrastingly, the daily cost for continuous intravenous sedatives was significantly lower in the EA group than in the NA group (β -17.6 [95% CI, -30.8 to -4.5])”

・“For all neuroactive drugs (SPM, sedatives, and analgesics), the daily costs tended to be lower in the EA group than in the LA and NA groups.” See previous comment.

We thank the reviewer for providing these insights. We have changed the sentence “For all neuroactive drugs (SPM, sedatives, and analgesics), the daily costs tended to be lower in the EA group than in the LA and NA groups.” to “For all neuroactive drugs (SPM, sedatives, and analgesics), the daily costs was significantly lower in the EA group than in the NA group (β -16.9 [95% CI, -31.4 to -2.5]).”

TABLE 1

・“Mechanical ventilation” does not seem to be reported in the table and should be removed from the caption.

Response: We have removed “MV: Mechanical ventilation” from the caption.

・P-values should be reported to test each qualitative or quantitative variable difference among the groups.

Response: The reviewer has raised a relevant point; however, as previously mentioned, we decided not to include P-values.

TABLE 2

・P-values should be reported to test each qualitative or quantitative variable difference among the groups.

Response: The reviewer has raised a relevant point; however, as previously mentioned, we decided not to include P-values.

TABLE 3

・“MV through an oral ET tube”: I would change to “MV duration through an oral ET tube”.

Response: We have modified the phrase according to the reviewer’s comment.

DISCUSSION

・“Regarding the costs for all neuroactive drugs, the EA group showed an advantageous tendency compared with the other groups”. The trend could be observed for the EA vs. LA comparison, but it passed the threshold for statistical significance for the EA vs. NA comparison. Provided that statistical analysis may need to be reviewed, this should be accounted for in the discussion and conclusions.

Response: We thank the reviewer for providing these insights. We have changed the sentence “Regarding the costs for all neuroactive drugs, the EA group showed an advantageous tendency compared with the other groups.” to “Regarding the costs for all neuroactive drugs, the EA group was significantly more advantageous than the NA group.”

・The choice of SPM was discretional and based on the authors’ own experience. This should be emphasized in the limitations section.

Response: We agree with the reviewer’s assessment. We have incorporated the comments by adding the following sentence in the limitation section: “Fourth, the choice of the SPM was based on the authors’ own experience, and it is unclear whether other drugs would have the same efficacy.”

SUPPLEMENTARY

・Supplementary materials should all be referred to in the main text.

Response: 

We have rewritten the supporting information citation in the manuscript according to the Supporting Information guidelines. In the study design section in Materials and Methods, we have changed “S1 Table” to “Table in S1 File”, and we have changed “S2 Table” to “Table in S2 File” in Outcomes in the Results section.

Reviewer #2

I would only suggest some corrections to improve fluency.

Here are a few examples:

Introduction

- Second paragraph: ARE administered. Please change

- Second paragraph: Please change “which cause” to “causing” in both instances to improve readability

- “Currently, there have been no studies on”. Please change to “To the best of our knowledge, no studies currently exist addressing…”

Methods

- Study design: “…study employed medical record information “. Please changed to “implied a revision of medical records”

Response: We thank the reviewer for providing these insights. We have revised the sentences according to the reviewer’s comments.

Reviewer #3

1- Clarify if patients were divided randomly into the three groups.

Response: We thank the reviewer for providing these insights. This study was a retrospective study; hence, the assignment to the three groups was not random. We believe that the word “random” is not appropriate for this study; thus, we have decided not to state whether the patients were randomly divided into three groups or not.

2- Indicate if the data was checked for normal distribution or transformed to a normal distribution when using multiple linear regression.

Response: The reviewer has raised a relevant question. We performed residual analysis using normal quantile-quantile (Q-Q) plot to check for normal distribution. We have incorporated the comments by adding the following sentence in the Statistical Methods section: “We checked residuals in multiple liner regression for normal distribution using normal quantile-quantile plot.”

We have added a new file named “S4 File” to show the normal Q-Q plot.

3- Page 7: Indicate the statistical testing method(s) used to conclude, “There were no among-group differences . . .”

Response: The reviewer raised a significant question. As mentioned in our response to Reviewer 1, we decided not to include p values in comparing baseline characteristic. Thus, we have modified the sentence that you pointed out: “There were no among-group differences in the age, sex, body mass index, SOFA scores on admission, Charlson comorbidity index, and use of sleeping pills before ICU admission.” to “The median age of patients of this study was 72 years, 72% were males, and the median body mass index was 23.1. The SOFA score and Charlson comorbidity index were similar in each group.”

4- Table 1: Replace “man” with “male.”

Response: We have replaced “man” with “male.”

5- In the statistical methods section, clarify the model used for summarizing the average daily cost.

Response: We thank the reviewer for providing these insights. No special model was used to calculate the cost in this study. We have incorporated the comments by adding the following sentence in the Statistical Methods section: “We did not use any special model to calculate the costs in this study. The cost was calculated by totaling the drugs used in this study.” 

6- Add line numbers to assist in the review process.

Response: We have added line numbers.

7- Carefully proofread the document. Use complete sentences and check grammar.

Response: I asked Editage (www.editage.com) to proofread the revised manuscript paper.

8- State and justify the study’s target sample size with a pre-study statistical power calculation. The power calculation should include: (1) the estimated outcomes in each group; (2) the α (type I) error level; (3) the statistical power (or the β (type II) error level); (4) the target sample size and (5) the statistical testing method and (6) for continuous outcomes, the standard deviation of the measurements.

Response: The reviewer has raised a significant point. Since our hospital had only been open for a short time, we included as many patients as we could. We believe that post hoc power calculations are futile for exploratory observational studies [1,2]. This is why we decided not to make the power calculation. 

[1] Vandenbroucke JP, et al; STROBE Initiative. Strengthening the Reporting of Observational Studies in Epidemiology (STROBE): explanation and elaboration. PLoS Med. 2007 Oct 16;4(10):e297. doi: 10.1371/journal.pmed.0040297.

[2] Lydersen S. Statistical review: frequently given comments. Ann Rheum Dis. 2015 Feb;74(2):323-5. doi: 10.1136/annrheumdis-2014-206186.

Again, thank you for giving us the opportunity to strengthen our manuscript with your valuable comments and queries. We have worked hard to incorporate your feedback and hope that these revisions persuade you to accept our submission.

---

## [Decision Letter · Decision Letter 1]

3 Nov 2021

PONE-D-21-05291R1Effect of enterally administered sleep-promoting medication on the intravenous sedative dose and its safety and cost profile in mechanically ventilated patients: A retrospective cohort study

PLOS ONE

Dear Dr. Tsunemitsu,

Thank you for submitting your manuscript to PLOS ONE. After careful consideration, we feel that it has merit but does not fully meet PLOS ONE’s publication criteria as it currently stands. Therefore, we invite you to submit a revised version of the manuscript that addresses the points raised during the review process.

Please modify the sentence according to Reviewer's #1 suggestion. I think it's more appropriate. 

We look forward to receiving your revised manuscript.

Kind regards,

Laura Pasin

Academic Editor

PLOS ONE

Journal Requirements:

Reviewers' comments:

Reviewer's Responses to Questions

**Comments to the Author**

1. If the authors have adequately addressed your comments raised in a previous round of review and you feel that this manuscript is now acceptable for publication, you may indicate that here to bypass the “Comments to the Author” section, enter your conflict of interest statement in the “Confidential to Editor” section, and submit your "Accept" recommendation.

Reviewer #1: All comments have been addressed

Reviewer #3: All comments have been addressed

2. Is the manuscript technically sound, and do the data support the conclusions?

Reviewer #1: Yes

Reviewer #3: (No Response)

3. Has the statistical analysis been performed appropriately and rigorously? 

Reviewer #1: I Don't Know

Reviewer #3: (No Response)

4. Have the authors made all data underlying the findings in their manuscript fully available?

Reviewer #1: Yes

Reviewer #3: (No Response)

5. Is the manuscript presented in an intelligible fashion and written in standard English?

Reviewer #1: Yes

Reviewer #3: (No Response)

6. Review Comments to the Author

Reviewer #1: Thank you for the opportunity to review this revised manuscript. I would like to thank the authors for addressing my concerns. No difference in primary and secondary outcomes between the groups when considered EA and LA together were reported. This was adequately commented on. I am still not sure about the adequacy of the statistical analysis. However, the authors stated they consulted a Statistician, who they should consider to acknowledge in the appropriate section. Please consider modifying the sentence “The SOFA score and Charlson comorbidity index were similar in each group”, since no statistical test for assessing among-group difference was performed, similar to the sentence regarding age, gender and BMI.

Reviewer #3: (No Response)

7. PLOS authors have the option to publish the peer review history of their article (what does this mean?). If published, this will include your full peer review and any attached files.

Reviewer #1: No

Reviewer #3: No

---

## [Author Response · Author response to Decision Letter 1]

5 Nov 2021

Emily Chenette

Editor-in-Chief

PLOS ONE 

Dear Editor,

Manuscript No: PONE-D-21-05291

Title: Effect of enterally administered sleep-promoting medication on the intravenous sedative dose and its safety and cost profile in mechanically ventilated patients: A retrospective cohort study

We would like to express our sincere thanks to you and the reviewers for the thorough review of our manuscript and for the opportunity to submit a revised and improved version. We have carefully reviewed the comments and revised the manuscript on the basis of the reviewers’ comments. Our point-by-point responses to the reviewers’ comments are listed below. 

Journal Requirements:

●Please review your reference list to ensure that it is complete and correct. If you have cited papers that have been retracted, please include the rationale for doing so in the manuscript text, or remove these references and replace them with relevant current references. Any changes to the reference list should be mentioned in the rebuttal letter that accompanies your revised manuscript. If you need to cite a retracted article, indicate the article’s retracted status in the References list and also include a citation and full reference for the retraction notice.

Response: I have checked the references and did not make any changes.

Reviewer #1: 

●Thank you for the opportunity to review this revised manuscript. I would like to thank the authors for addressing my concerns. No difference in primary and secondary outcomes between the groups when considered EA and LA together were reported. This was adequately commented on. I am still not sure about the adequacy of the statistical analysis. However, the authors stated they consulted a Statistician, who they should consider to acknowledge in the appropriate section. 

Response: We agree with the reviewer’s assessment. Dr. Yuki Kataoka is an expert in statistics, and He performed the statistical processing for this study. In the Authors' contributions section, We’ve already described his contribution to the statistical process.

●Please consider modifying the sentence “The SOFA score and Charlson comorbidity index were similar in each group”, since no statistical test for assessing among-group difference was performed, similar to the sentence regarding age, gender and BMI.

Response: We thank the reviewer for providing these insights. We have rewritten the sentences according to the reviewer’s comments: “The SOFA score and Charlson comorbidity index were similar in each group” to “The SOFA score and Charlson comorbidity index were 6 and 0 in the EA group, 6 and 0 in the LA group, and 7 and 0 in the NA group.”

Again, thank you for giving us the opportunity to strengthen our manuscript with your valuable comments and queries. We have worked hard to incorporate your feedback and hope that these revisions persuade you to accept our submission.

Sincerely,

Takefumi Tsunemitsu

Department of Emergency and Critical Care Medicine, Hyogo Prefectural Amagasaki General Medical Center 2-17-77 Higashinaniwa-cho, Amagasaki, Hyogo, 660-8550, Japan

Phone: 81-6-6480-7000

Fax: 81-6-6480-7001

E-mail: tsunemitsu0730@yahoo.co.jp

---

## [Editor Report · Decision Letter 2]

1 Dec 2021

Effect of enterally administered sleep-promoting medication on the intravenous sedative dose and its safety and cost profile in mechanically ventilated patients: A retrospective cohort study

PONE-D-21-05291R2

Dear Dr. Tsunemitsu,

We’re pleased to inform you that your manuscript has been judged scientifically suitable for publication and will be formally accepted for publication once it meets all outstanding technical requirements.

Kind regards,

Laura Pasin

Academic Editor

PLOS ONE
---

## [Editor Report · Acceptance letter]

6 Dec 2021

PONE-D-21-05291R2 

Effect of enterally administered sleep-promoting medication on the intravenous sedative dose and its safety and cost profile in mechanically ventilated patients: A retrospective cohort study 

Dear Dr. Tsunemitsu:

I'm pleased to inform you that your manuscript has been deemed suitable for publication in PLOS ONE. Congratulations! Your manuscript is now with our production department. 

Kind regards, 

on behalf of

Dr. Laura Pasin 

Academic Editor

PLOS ONE